# PHRF1 promotes migration and invasion by modulating ZEB1 expression

Jin-Yu Lee[ID][1☯], Chih-Chen Fan[2,3☯], Nai-Lin Chou[1], Hung-Wei Lin[1], Mau-Sun Chang[ID][1,4]*

**1** Institute of Biochemical Sciences, National Taiwan University, Taipei, Taiwan, **2** Department of Superintendent Office, Mackay Memorial Hospital, Taipei, Taiwan, **3** Department of Medical Laboratory Science and Biotechnology, Yuanpei University, Hsinchu, Taiwan, **4** Institute of Biological Chemistry, Academia Sinica, Taipei, Taiwan

☯ These authors contributed equally to this work.
* mschang@ntu.edu.tw

**Data Availability Statement:** All relevant data are within the paper and its Supporting Information files.

**Funding:** This work was supported by Ministry of Science and Technology (MOST 106-2311-B-002-

## Abstract

PHRF1 (PHD and RING finger domain-containing protein 1) suppresses acute promyelocytic leukemia (APL) by promoting TGIF (TG-interacting factor) ubiquitination, while the PML-RARα protein interferes with PHRF1-mediated TGIF breakdown to facilitate APL. Beyond its role in APL tumorigenesis, PHRF1 contributes to non-homologous end-joining by linking H3K36 methylation and Nbs1 upon DNA damage insults. However, little is known regarding its function in tumor invasion. Here we highlight the unreported details of PHRF1 in the invasion of lung cancer cells by modulating the transcriptional level of *ZEB1*, a prominent regulator involved in epithelial-mesenchymal transition. PHRF1 associated with the phosphorylated C-terminal repeat domain of Rpb1, the large subunit of RNA polymerase II, through its C-terminal Set2 Rpb1 Interacting (SRI) domain. Chromatin immunoprecipitation revealed that PHRF1 bound to the proximal region adjacent to the transcription start site of *ZEB1*. SRI-deleted PHRF1 neither associated with Rpb1 nor increased *ZEB1*'s expression. Collectively, PHRF1 might take the stage at migration and invasion by modulating the expression of ZEB1.

## Introduction

Metastasis is a complex and multistep process by which cancer cells enable primary tumors to invade the surrounding stroma, travel through the vasculature, and then colonize in distant tissues [1]. Mounting evidence embraces the fact that major players involved in the epithelial-mesenchymal transition (EMT) promote mobility for cancer cells to generate migration and invasion in solid tumors. The EMT, which converts epithelial cells into motile and invasive mesenchymal phenotypes, potentiates the dissemination of tumor cells during metastasis [2–5]. The EMT switch is initiated by several key transcription factors, including SNAI1, ZEB1/ZEB2 (Zinc finger E-box-binding homeobox 1/2) and Twist. Consequently, loss of E-cadherin and upregulation of N-cadherin have been frequently identified during EMT. Among these transcription factors, ZEB1 and ZEB2 are triggered by multiple signaling cascades such as TGF-β and HIF-1 (hypoxia inducing factor-1) [6, 7], which are tumor-bearing

004). The funders had no role in study design, data collection and analysis, decision to publish, or preparation of the manuscript. All authors did not receive a salary from our funders.

**Competing interests:** Additionally, the authors have declared that no competing interests exist.

microenvironment factors to promote EMT and metastasis. Additionally, ZEB1 and ZEB2 overexpression have been found in several human cancers, including non–small cell lung cancer (NSCLC) [8, 9]. In lung cancer cell lines, ZEB1 is inversely correlated with expression of E-cadherin and promotes anchorage-independent colony formation [10, 11]. More recently, Larsen *et al.* presented a model regarding ZEB1-induced EMT for malignant transformation, which is an early and critical event in lung cancer pathogenesis [12].

Rpb1, the largest subunit of RNA polymerase II complex (RNAPII), contains approximately 52 tandem repeats of heptapeptide YSPTSPS in its C-terminal repeat domain (CTD). The phosphorylated CTD serves as a flexible binding scaffold for numerous nuclear factors to facilitate transcription progression [13, 14]. Three serines (Ser2, Ser5, Ser7), one tyrosine (Tyr1) and one threonine (Thr4) can be phosphorylated during different transcriptional stages [15, 16]. Successively, RNAPII is recruited to promoters in a non-phosphorylated form, but becomes phosphorylated on Ser5 and Ser7 prior to initiation. After transcribing 20–60 nucleotides, RNAPII pauses downstream of the transcription start site (TSS) upon the binding of the DRB sensitivity-inducing factor (DSIF) and Negative Elongation Factor (NELF). A wealth of factors has been known to participate in transcriptional elongation [17]. Particularly, the von Hippel-Lindau tumor suppressor protein (pVHL) inhibits transcription elongation by competing against Elongin A to associate with Elongin B and C [18].

Human PHRF1 contains a plant homeodomain (PHD) that recognizes methylated histones and a RING domain which ubiquitinates substrates. Structurally, the C-terminus of PHRF1 harbors an SRI domain that is predicted to interact with the phosphorylated CTD of Rpb1 [19]. Previous studies have shown that PHRF1 promotes TGF-β signaling through TGIF ubiquitination to ensure redistribution of cPML (the cytoplasmic variant of promyelocytic leukemia protein) to the cytoplasm, where cPML associates with SARA (Smad anchor for receptor activation) to coordinate the activation of Smad2 by TGF-β receptors [20]. Abnormal PML-RARα fusion protein competes with PHRF1's binding to TGIF and interferes the TGIF breakdown. Consequently, cPML is ubiquitinated by TGIF [21]. We report a different aspect of PHRF1 in modulating non-homologous end-joining (NHEJ) in which PHRF1 combines with H3K36 methylation and NBS1 (the Nijmegen breakage syndrome gene) and a component of the MRE11/RAD50/NBS1 (MRN) complex, to stabilize genomic integrity by promoting PARP1 polyubiquitination for proteasomal degradation [22]. Recently, Wang *et al.* described that PHRF1 may attenuate the proliferation and tumorigenicity of non-small cell lung cancer cells. A lower level of PHRF1 mRNA was observed in human lung cancer tissues than in paracancerous tissues. Overexpression of PHRF1 arrested the cell cycle in the G1 phase and inhibited H1299 cell proliferation, colony formation *in vitro*, and growth of tumor xenograft *in vivo* [23].

Here we report that PHRF1 overexpression facilitated EMT in human lung cancer A549, CL1-0, and CL-1-5 cells, as shown in enhancement of migration, and invasion *in vitro* and *in vivo*. These functional effects of PHRF1 on EMT and invasion were largely dependent on ZEB1. We also provide evidence that PHRF1 controls the expression of *ZEB1* by collaborating with RNAPII. Together, our findings cement a novel link of PHRF1 with ZEB1 in the process of lung cancer metastasis.

## Materials and methods

### Cell lines

A549 human lung cancer cells obtained from the American Type Culture Collection (ATCC; Rockville, MD) and cultured in F12K medium (Hyclone, Utah, USA) supplemented with 10% FBS. HEK293T cells were obtained from ATCC and maintained in DMEM medium (Hyclone,

Utah, USA) with 10% FBS. Human lung adenocarcinoma cells CL1-0 and CL1-5 were established at The National Taiwan University College of Medicine and maintained in RPMI1640 medium (Hyclone, Utah, USA) with 10% FBS. All cell lines were submitted to real time PCR for mycoplasma detection and short tandem repeats identification by capillary electrophoresis for cell line authentication.

## Antibodies

The mouse anti-PHRF1 monoclonal antibody has been described [18]. Mouse anti-HA were purchased from Santa Cruz Biotech (Cat. No. SC-7392, Santa Cruz, CA). Rabbit anti-ZEB1 (Cat. No. 105278), ZEB2 (Cat. No. 129243), E-cadherin (Cat. No. 100443), and N-cadherin (Cat. No. 127345) antibodies were obtained from GeneTex (Hsinchu, Taiwan). Anti-Rpb1 (Cat. No. 2629), anti-Rpb1 CTD Ser2 (Cat. No. 13499) and anti-Rpb1 CTD Ser5 (Cat. No. 13523) antibodies were from Cell Signaling (Danvers, MA). Mouse anti-α-tubulin antibody was from Santa Cruz Biotech (Cat. No. SC-32293, Santa Cruz, CA). Mouse anti-β-actin antibody was from Novus (Cat. No. NB-600501, Littleton, CO).

## RNA interference

The short hairpin RNA (shRNA) shPHRF1#1, 5'-CCTGTGTTGCTCACAGTTGAA-3' and shPHRF1#2, 5'-CGGACACGTCTTTGATGATTT-3' were obtained from RNAi Core Facility (Academia Sinica, Taipei, Taiwan).

## Quantitative PCR

Total RNAs were prepared using the RNAeasy kit (Qiagen). Quantitative PCR (qPCR) was performed using the SYBR Green Master Mix (Protech Technology, Taipei, Taiwan) on the ABI 7300 Real-time PCR System (Applied Biosystems, CA). The reaction contained 2 μl of cDNA and 0.2 μM of primers in a final volume of 20 μl master mix. Forty cycles consisted of denaturation at 95˚C for 15 s and annealing at 60˚C for 60 s. GAPDH was used as an endogenous control to normalize each sample. The primers are listed in the S1 Table. Three independent experiments were performed.

## Chromatin immunoprecipitation (ChIP) PCR

We used the EpiTect ChIP OneDay kit (Qiagen, Germantown, MD) for ChIP-PCR. Briefly, control and HA-PHRF1 overexpressing A549 cells were fixed by formaldehyde (1% final concentration) for 10 min at 37˚C and quenched by glycine (final concentration 0.125 M). Cells were harvested in RIPA buffer and sonicated to reduce the DNA length. The soluble fraction was diluted in ChIP dilution buffer and complexes were pulled down by anti-HA agarose, cell lysates at 4˚C overnight (Sigma-Aldrich). The bound immunocomplex then reversed the cross-linking and was treated with proteinase K. DNA was isolated and then subjected to PCR. The primers for PCR are listed in the S2 Table.

## Cell migration assay

$1x10^4$ A549 cells and its derivative cell lines were seeded in 6-well plates until 95% confluence. The wound was created by pipette tip across the monolayer. Cells were washed with PBS and placed in the fresh culture medium containing 5% FBS. The cell motility was recorded under microscope every 6 h.

## Cell invasion assay

Twenty-four hours after transfection, matrigel invasion assays were conducted using 8 mm Transwell chambers. Matrigel was diluted in cold distilled water, added to the upper wells of the Transwell chambers (2 mg/well), and dried in a sterile hood. Matrigel was reconstituted with medium for 3 h at 37°C before the addition of cells. Cells were resuspended in serum-free medium containing 0.1% FBS at a concentration of $1x10^4$ cells with 300 μl of serum-free medium seeded into the upper chamber. 700 μl of medium containing 10% FBS was added into the lower wells of the 24-well plate. After 24 h for invasion assays, cells on the underside of the membrane of chambers were fixed in methanol and then stained with crystal violet. Invaded cells were recorded under a microscope and counted with three independent experiments.

## *In vivo* ubiquitination

HEK293T cells were co-transfected with FLAG-tagged ubiquitin (Ub), HA-tagged PHRF1$^{WT}$ or PHRF1 RING mutant (HA-PHRF1$^{C108A}$), and Streptavidin binding peptide (SBP)-tagged pVHL with or without the addition of MG132 (10 μM, Sigma-Aldrich) for 3 h. Total cell extracts were harvested with RIPA buffer at 48 h post transfection for immunoblotting analysis.

## Animal studies

All animal studies were performed in compliance with the guidelines of the Institutional Animal Care and Use Committee at Academia Sinica with the approved protocol no. 15-11-184. Three mice of each group were purchased from BioLASCO (Taipei, Taiwan). All animals received standard care, including free access to food and water, a 12/12 light/dark cycle, and constant temperature and humidity. $5x10^4$ A549 cells were inoculated into the tail veins of SCID mice. Mice were sacrificed in a $CO_2$ chamber at three weeks after inoculation. To prevent a possibility that mice might recover from a deep $CO_2$ exposure, a confirmatory means of euthanasia, such as cervical dislocation or 50% additional time in the euthanasia chamber, was conducted. Daily monitoring was conducted to ensure there were no adverse effects due to tumor inoculations. The well being of the animals had priority over the continuation of planned procedures. All procedures were performed to minimize pain and distress. Developing tumors might result in some levels of distress or discomfort in these mice. If there were any signs of post-procedure pain, such as 20% loss of body weight, tumors grown over 10% of body mass, changes in behavior, inactivity, prostration, poor breathing, hunched posture, skin ulcers and abnormal vocalization, the animal was immediately euthanized in a $CO_2$ chamber.

## Immunohistochemistry (IHC)

A study regarding "the expression of PHRF1 in the lung carcinoma" was approved by the Institutional Review Board of Mackay Memorial Hospital (IRB No: 15MMHIS012) and the methods were carried out in accordance with the approved guidelines. Briefly, signed informed consents were obtained from all participants. Tissue specimens of 80 patients with lung cancer carcinomas, ranging from January 2008 to June 2014, were selected for IHC analyses based on availability of archival human lung tissue blocks from diagnostic inspections by pathologists at MacKay Memorial Hospital, Taipei, Taiwan. Paraffin sections (4 mm) were deparaffinized in xylene solution and rehydrated through graded alcohols, followed by heat-induced antigen retrieval in citric acid buffer (pH 6.0) for 20 min. Endogenous peroxidase activity was then blocked with 0.5% $H_2O_2$ in methanol for 30 min. Sections were incubated with indicated

antibodies and developed with the DAB Chromogen Kit (Biocare). Sections were then counterstained with hematoxylin and scored (zero, no staining; one, weak; two, moderate; three, strong staining) under light microscope.

## Statistical analysis

Analysis was carried out using GraphPad Prism 6 software. All values were expressed as mean ± SD. The paired Student's $t$-test (two-tailed) was used to calculate the statistical significance of differences between groups. The $p < 0.05$ was considered statistically significant. The Spearman test was used to analyze the correlation for IHC results in clinical specimens.

# Results

## PHRF1 expression is associated with overall and progression free survival

To investigate the potential role of PHRF1 in cancer progression, we examined the association of PHRF1 expression level with overall and progression-free survival in a number of cancer patients. Using the Kaplan Meier Plotter (http://kmplot.com/analysis/index.php?p= service&cancer) [24], a publicly accessible database, a high level of *PHRF1* is notably associated with poor survival in the ovarian and gastric mRNA database. Further supporting this finding, the high level of *PHRF1* is associated with poor survival in mRNA-seq analysis from lung squamous carcinoma, liver hepatocellular carcinoma, and cervical cell carcinoma (S1A–S1F Fig). Additionally, an elevated *PHRF1* expression is associated with poor survival in the lung adenocarcima and cervical squamous cell carcinoma using the cBioportal TCGA dataset analysis (S1G and S1H Fig), indicating that PHRF1 expression is associated with survival in some kinds of cancer patients.

## PHRF1 affected transwell migration and invasion

To illustrate whether PHRF1 affected cell mobility, migration assays using wound healing and Boyden's transwell were conducted. Interestingly, knockdown of PHRF1 impeded cell mobility both in wound healing and transwell assays compared with control cells (Fig 1A and 1B). Conversely, overexpression of PHRF1 accelerated cell migration (Fig 1C and 1D). Since epithelial-mesenchymal transition (EMT) is considered a key factor for cell migration, the expression levels of EMT markers, such as E-cadherin and N-cadherin, were examined in PHRF1-depleted and -overexpressing cells. As anticipated, elevated E-cadherin and decreased N-cadherin were found in PHRF1-depleted cells. In contrast, compromised E-cadherin and was identified in PHRF1-overexpressing cells (Fig 1E), indicating that PHRF1 might modulate the expressing of the EMT's components to affect cell migration.

To verify whether PHRF1 potentiates cell mobility, a transwell invasion assay mixed with Matrigel to mimic the environment for invasion was conducted. The majority of control A549 cells penetrated the Matrigel at 24 h; nonetheless, most of PHRF1-depleted cells were unable to move to the lower membrane (Fig 2A). By contrast, a large number of PHRF1-overexpressing cells invaded to the lower environment compared with control cells (Fig 2B), indicating that PHRF1 can promote invasion in Matrigel. Next, to uncover the effect of PHRF1 on tumor metastasis *in vivo*, PHRF1-depleted and -overexpressing A549 cancer cells were inoculated into the tail veins of SCID mice. Mice injected with control cells displayed robust lung tissues with little amount of cell aggregateion. However, the injection of PHRF1-overexpressing A549 cells resulted in remarkably enlarged lung tissues with visible aggregates atthree-weeks post inoculation. Histological examinations revealed that PHRF1-overexpressing A549 cells expanded in the entire alveolar spaces (Fig 2C). To a much lesser extent, control A549 cells

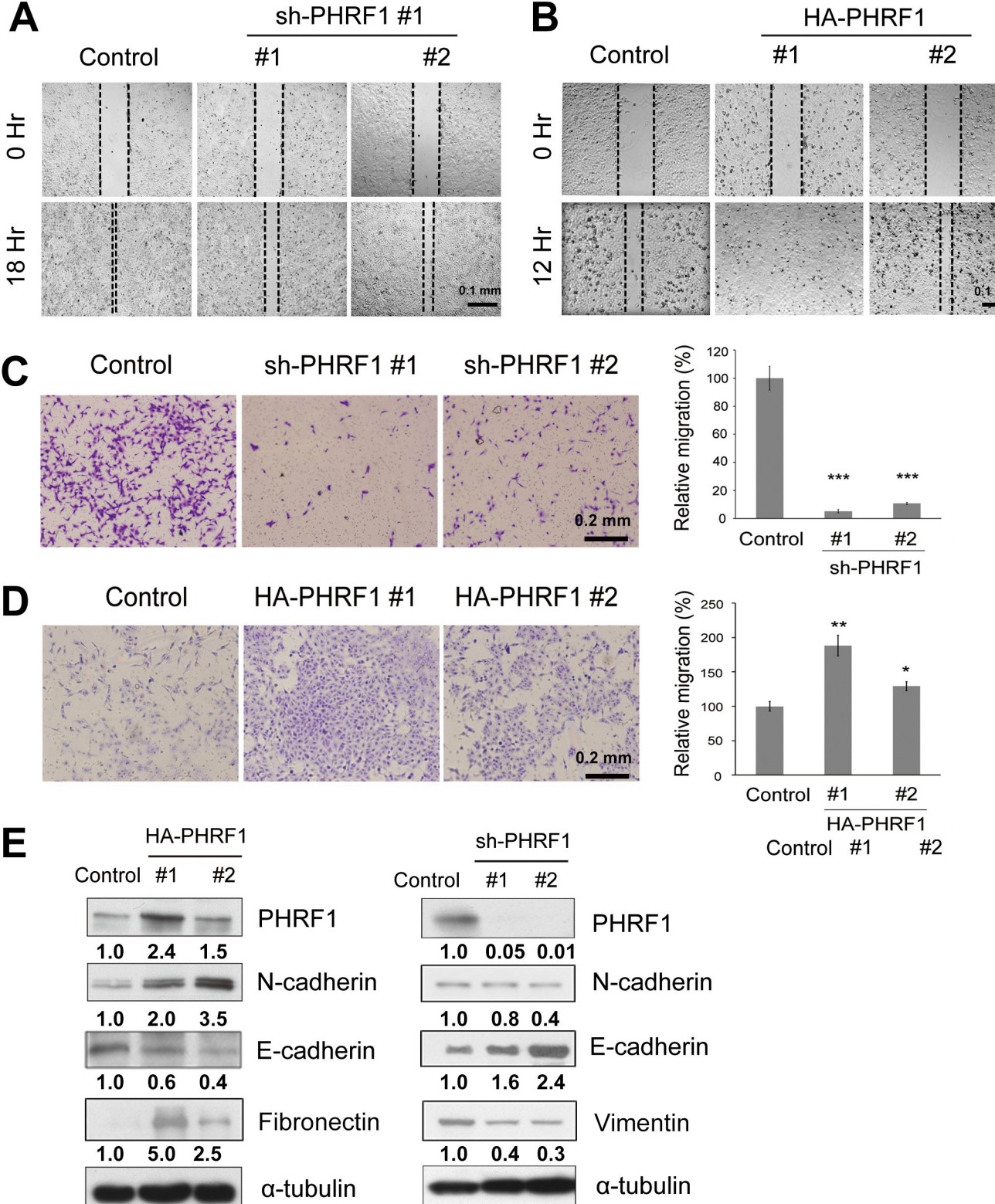

**Fig 1. Effects of PHRF1 on cell migration.** (A, B) PHRF1-depleted and -overexpressing A549 cells were subjected to wound healing assay for 20 or 14 h, respectively. Scale bar, 0.1 mm. (C, D) PHRF1-deleted and -overexpressing A549 cells were seeded on the top of a Boyden chamber for migration assay. Cells were allowed to migrate to the lower chamber for 24 h. Cells moved to the lower membranes were photographed. Relative transwell migration was quantified compared with the control. Each bar represents the mean ± SD of three independent experiments. ($^*$P < 0.05, $^{**}$P < 0.01, $^{***}$P < 0.001). Scale

bar, 0.2 mm. (E) Immunoblotting analyses were conducted against cell extracts prepared from PHRF1-overexpressing and -depleted A549 cells. All Western blots were processed in identical conditions and cropped from S1 Raw images.

formed smaller nodules in the spaces of lung epithelia. By contrast, PHRF1 depletion resulted in an opposite effect on metastasis (Fig 2D). Quantitative data showed that tumor volume was significantly increased in PHRF1- overexpressing cells compared with control cells (Fig 2E).

To unravel whether PHRF1 contributed to clonogenic formation, we silenced PHRF1's expression in A549 cells and examined its colony-forming capability. The number of colonies was not significantly changed in PHRF1-depleted or -overexpressing A549 cells compared with control cells. Nevertheless, most of PHRF1-depleted A549 cells formed smaller colonies (<1 mm) (S2A Fig). As a corollary, PHRF1-overexpressing A549 cells displayed larger colonies (>1 mm) compared with control cells (S2B Fig). The difference in colony size might be due to different proliferation capabilities, since overexpression of PHRF1 enabled cells with faster proliferation rates (S2C Fig). To clarify whether proliferation rate affected the cell migration, PHRF1-depleted and -overexpressing A549 cells were cultured in 0.5% of FBS and wound healing assays were conducted for 24 hr. PHRF1 knockdown significantly reduced the motility, similar to normal serum condition. Instead, PHRF1 overexpression had little effect on promoting cell migration compared with controls in 0.5% of FBS medium (S2D Fig), indicating that the cell motility of PHRF1 overexpression might be affected in low proliferation condition.

## ZEB1 is responsible for PHRF1-mediated migration and invasion

ZEB1 is a prominent transcription regulator involved in EMT and metastasis. In addition to A549 cells, ZEB1 was also elevated in PHRF1-transfected lung adenocarcinoma CL1-0 and CL1-5 cells (Fig 3A). Similarly, transwell migration and invasion were suppressed in PHRF1-depleted CL1-5, but increased in PHRF1-overexpressing CL1-0 cells (S3 Fig), indicating that PHRF1 was reliant on ZEB1 to induce migration and invasion. To confirm this speculation, two ZEB1 shRNAs were utilized to knockdown ZEB1's expression in PHRF1-overexpressing A549 cells (Fig 3B). Subsequent transwell migration and invasion assays revealed that knockdown of ZEB1 considerably compromised the invasive capability of PHRF1-overexpressing cells (Fig 3C), supporting the notion that ZEB1 virtually contributed to PHRF1-mediated EMT events.

## PHRF1 associates the genomic region of *ZEB1* adjacent to the Transcription Start Site (TSS)

To disclose the mechanism in which PHRF1 is accountable for the induction of ZEB1's expression, several PHRF1 deletion mutants were transfected into CL1-0 cells to measure the amount of ZEB1 by immunoblotting analysis. E3 ligase mutant (PHRF1$^{C108A}$) and PHD-deleted PHRF1 mutant (PHRF1$^{\Delta PHD}$) acted similarly as wild-type PHRF1, which increased the expression of ZEB1. By contrast, SRI-deleted PHRF1 (PHRF1$^{\Delta SRI}$, an interaction domain with the CTD of Rpb1) did not upregulate the expression of ZEB1 (Fig 4A), suggesting that ZEB1 was regulated by the SRI domain of PHRF1 to work with RNAPII. Because the SRI domain of PHRF1 is predicted to bind to the phosphorylated CTD of Rpb1 [19], immunoprecipitation was performed to validate whether PHRF1 engaged with Rpb1 by the SRI domain. As anticipated, PHRF1 was able to pull down Rpb1. By contrast, PHRF1$^{\Delta SRI}$ did not form an immuno-complex with Rpb1 (Fig 4B). Moreover, HA-PHRF1 predominantly immunoprecipitated the CTD of RNAPII phosphorylated on S2 and S2/S5 residues, but with little amount of the CTD

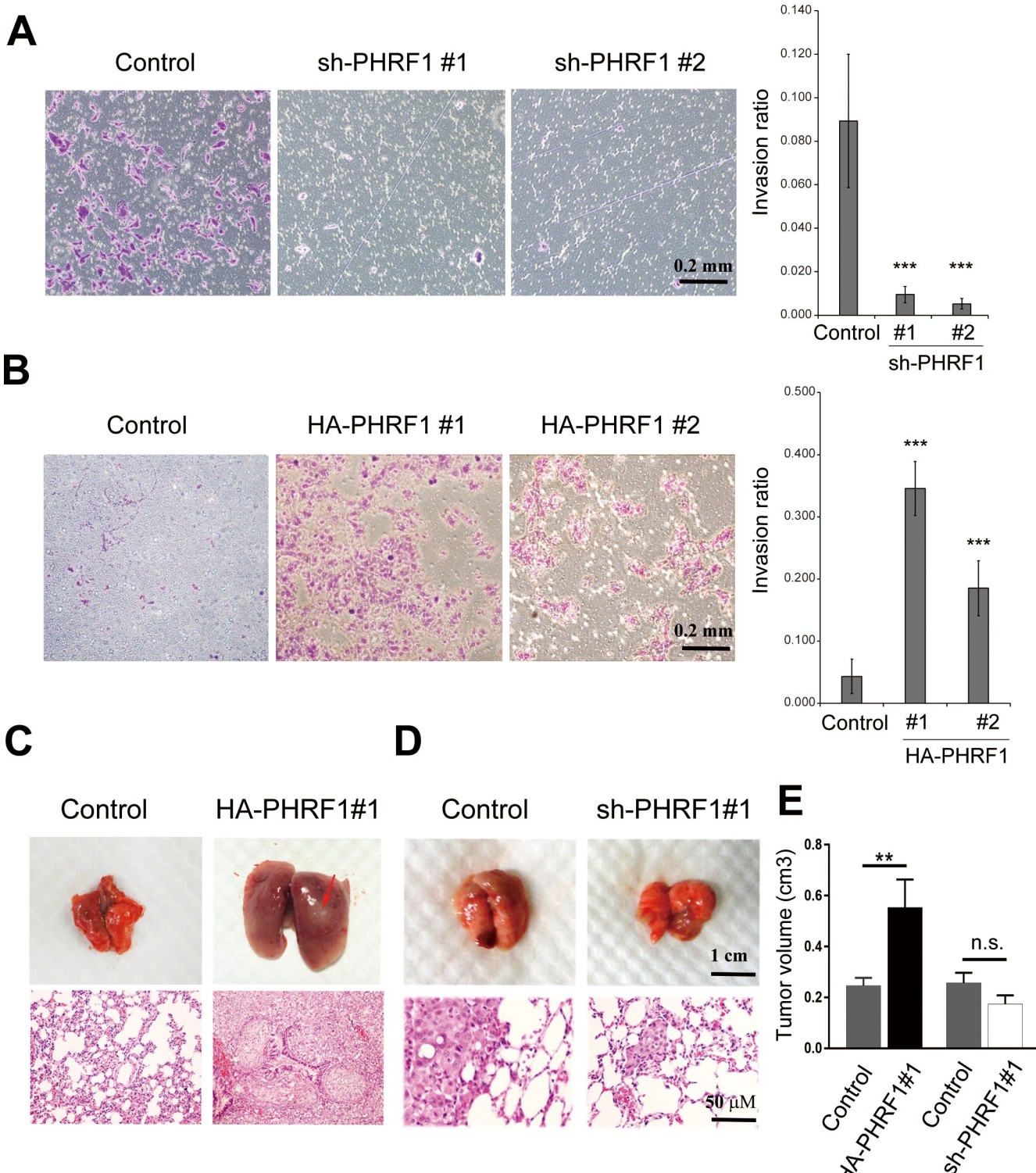

**Fig 2. PHRF1 promotes cell invasion *in vitro* and tumor metastasis *in vivo*.** (A, B) PHRF1- depleted and -overexpressing A549 cells were loaded onto the Matrigel for invasion assay. Cells penetrating to the lower surface of the membrane were stained with crystal violet and photographed. Invasion ratio was determined by the number of migrated cells in a confined area (***P < 0.001). Scale bar, 0.2 mm. (C, D) Control and PHRF1-overexpressing or -depleted A549 cells were injected into the tail veins of three SCID mice. Mice were sacrificed at 3 weeks after inoculation. Metastatic nodules were indicated by arrowheads. Scale bar, 1 cm in top panel; 50 μm in lower panel. (E) Tumor volumes were calculated using the formula (width$^2$) x length/2. Each bar represents the mean ± SD of three independent experiments. **P < 0.01.

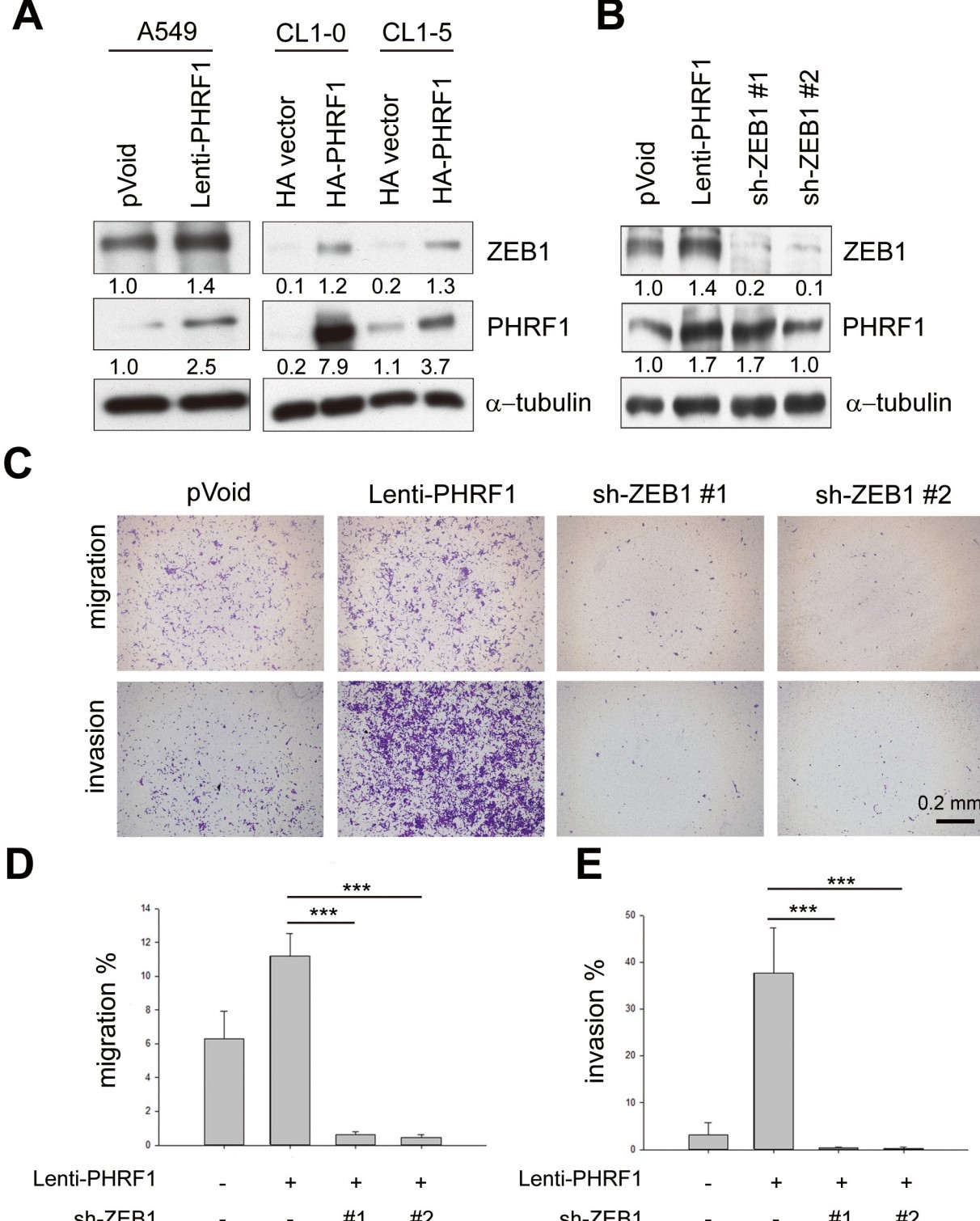

**Fig 3. ZEB1 is required for PHRF1-driven invasion.** (A) Cell extracts prepared from PHRF1- overexpressing A549 cells and PHRF1-transfected CL1-0, and CL1-5 lung cancer cells were immunoblotted with indicated antibodies. (B) PHRF1-overexpressing A549 cells were transduced with ZEB1 shRNAs and immunoblotting analysis was conducted with indicated antibodies. All Western blots were processed in identical conditions and cropped from S1 Raw images. (C) Transwell migration and Matrigel invasion were monitored in ZEB1-depleted PHRF1-overexpressing A549 cells. Scale bar, 0.2 mm. (D) Quantification of migration and invasion in ZEB1-depleted PHRF1-overexpressing A549 cells. All experiments were repeated three times. Error bars represent mean ± SD (***$P < 0.001$).

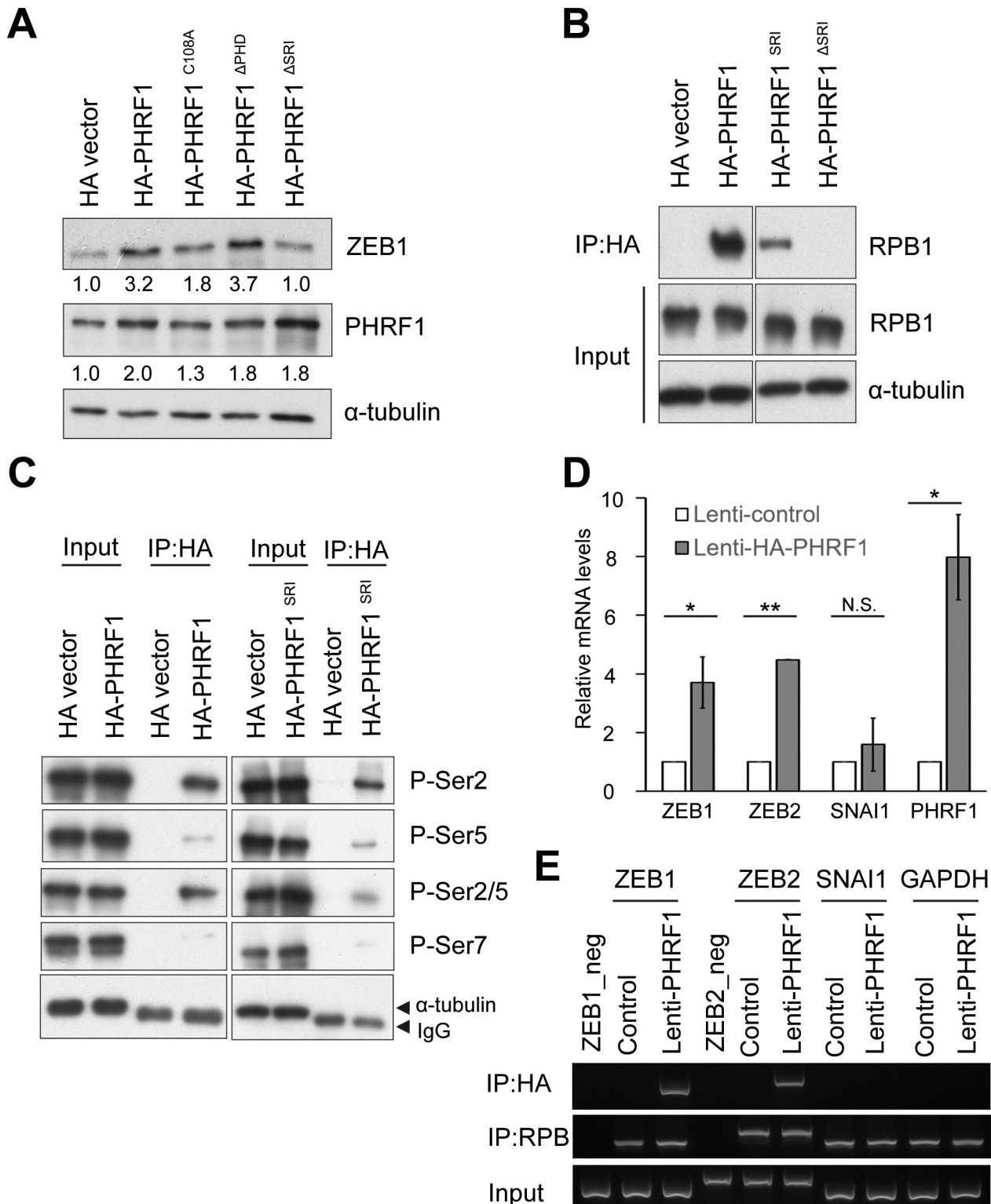

**Fig 4. PHRF1 modulates the transcription of *ZEB1*.** (A) Control and PHRF1 aberrant mutants were transfected into A549 cells and immunoblot analysis was carried out with indicated antibodies. (B) HA vector, HA-PHRF1, HA-PHRF1^SRI, and HA-PHRF1^ΔSRI were transfected into HEK293T cells, immunoprecipitated with anti-HA-beads, and immunoblotted with indicated antibodies. (C) HA-PHRF1 and HA-PHRF1^SRI were transfected into HEK293T cells, immunoprecipitated with anti-HA-beads, and immunoblotted with a variety of anti-phosphorylated CTD antibodies. All Western blots were processed in identical conditions and cropped from S1 Fig. (D) Total RNAs were prepared from control and HA-PHRF1-

overexpressing A549 cells for quantitative RT-PCR (qPCR). All experiments were normalized with *GAPDH* mRNA and repeated three times independently. Error bars, mean ± SD, $^*P < 0.05$, $^{***}P < 0.001$. (E) Control and HA-PHRF1-overexpressing A549 cells were fixed with paraformaldehyde and immunoprecipitated for chromatin immunoprecipitation. ChIP-PCR was conducted using indicated primers. *ZEB1*_neg and *ZEB2*_neg are DNA fragments located at 1 kb upstream from their corresponding TSSs and immunoprecipitated with indicated antibodies using HA-PHRF1-overexpressing A549 cell extracts. All gel images were processed in identical conditions and cropped from S2 Raw images. Note that RPB1 was as a control.

phosphorylated on S5 and S7 (Fig 4C), indicating that PHRF1 might act through the transcription initiation and elongation complex to promote ZEB1's expression. Indeed, quantitative RT-PCR (qPCR) showed that PHRF1 specifically increased the transcription of *ZEB1* and *ZEB2*, but not *SNAI1* (Fig 4D). Since RNAPII paused at the +20-+60 downstream of the TSS before elongation proceeded, we designed a pair of primers surrounding the TSS of *ZEB1* and *ZEB2* for ChIP-PCR. The result showed that PHRF1 bound to the TSS regions of *ZEB1* and *ZEB2*, but not to 1 kb upstream of TSS (ZEB1-neg & ZEB2-neg) and those of *SNAI1* and *GAPDH* (Fig 4E), suggesting that PHRF1 was recruited to the TSS of *ZEB1*/*ZEB2*, possibly thereby facilitating their transcription.

## PHRF1 is required for pVHL ubiquitination

Hypoxia is a potent microenvironmental cue to promote the expression of ZEB1 in metastatic progression [25]. We analyzed the promoter sequences of PHRF1 with bioinformatics, and found that there were two putative hypoxia response elements (HRE1 CTACGTG and HRE2 TGACGTA) located at -363 to -369 nt and -293 to -299 nt upstream of *PHRF1*'s TSS (S4A Fig), indicating that *PHRF1* might be a downstream target regulated by HIF upon hypoxia. Nevertheless, PHRF1 remained unchanged when HIF1α and HIF2α were transfected into HEK293T cells (S4B Fig), militating against the possibility that HIF1α and HIF2α enhanced the expression of *PHRF1* under hypoxia.

To elaborate whether PHRF1 affected the expression of ZEB1 under hypoxia, control and PHRF1 knockdown A549 cells were exposed to hypoxia (1% $O_2$) and immunoblotting analyses were carried out. Evidently, PHRF1 and ZEB1 were induced in control cells under hypoxia; nonetheless, the amount of ZEB1 was unable to increase in PHRF1-depleted cells under low $O_2$ conditions (S5A Fig). Instead, we noticed that the amounts of HIF1α and HIF2α were decreased in PHRF1-depleted cells under hypoxia conditions (S5A Fig). Initially, we thought that PHRF1 might bind to the TSS of HIF1α and HIF2α to modulate their transcription. However, an intriguing possibility emerged whether PHRF1 targeted pVHL, a substrate recognition subunit of E3 ligase for the destruction of HIF1α and HIF2α under normoxia [26], for degradation. The amount of pVHL in PHRF1 aberrant mutant transfected A549 cells was measured by immunoblotting analysis. Only the E3 ligase mutant (PHRF1[C108A]) did not downregulate the expression of pVHL (S5B Fig), indicating that pVHL was regulated via the Ring domain of PHRF1. Furthermore, the amounts of pVHL were increased in PHRF1-depleted cells under hypoxia conditions (S5C Fig). Finally, pVHL was polyubiquitinated and accumulated in the presence of PHRF1 and proteasome inhibitor MG132. Polyubiquitinated pVHL, reduced in PHRF1[C108A] transfected cells and ubiquitin[K48R], was unable to incorporate into pVHL in the presence of PHRF1 (S5D Fig), supporting the notion that PHRF1 is responsible for the ubiquitination of pVHL *in vivo*.

## Positive co-expression of PHRF1 and ZEB1 in lung cancer specimens

Finally, we examined the expression of PHRF1 and ZEB1 in human lung cancer specimens (Fig 5A). Invariably, moderate and strong expression of ZEB1 was found in 60% of PHRF1

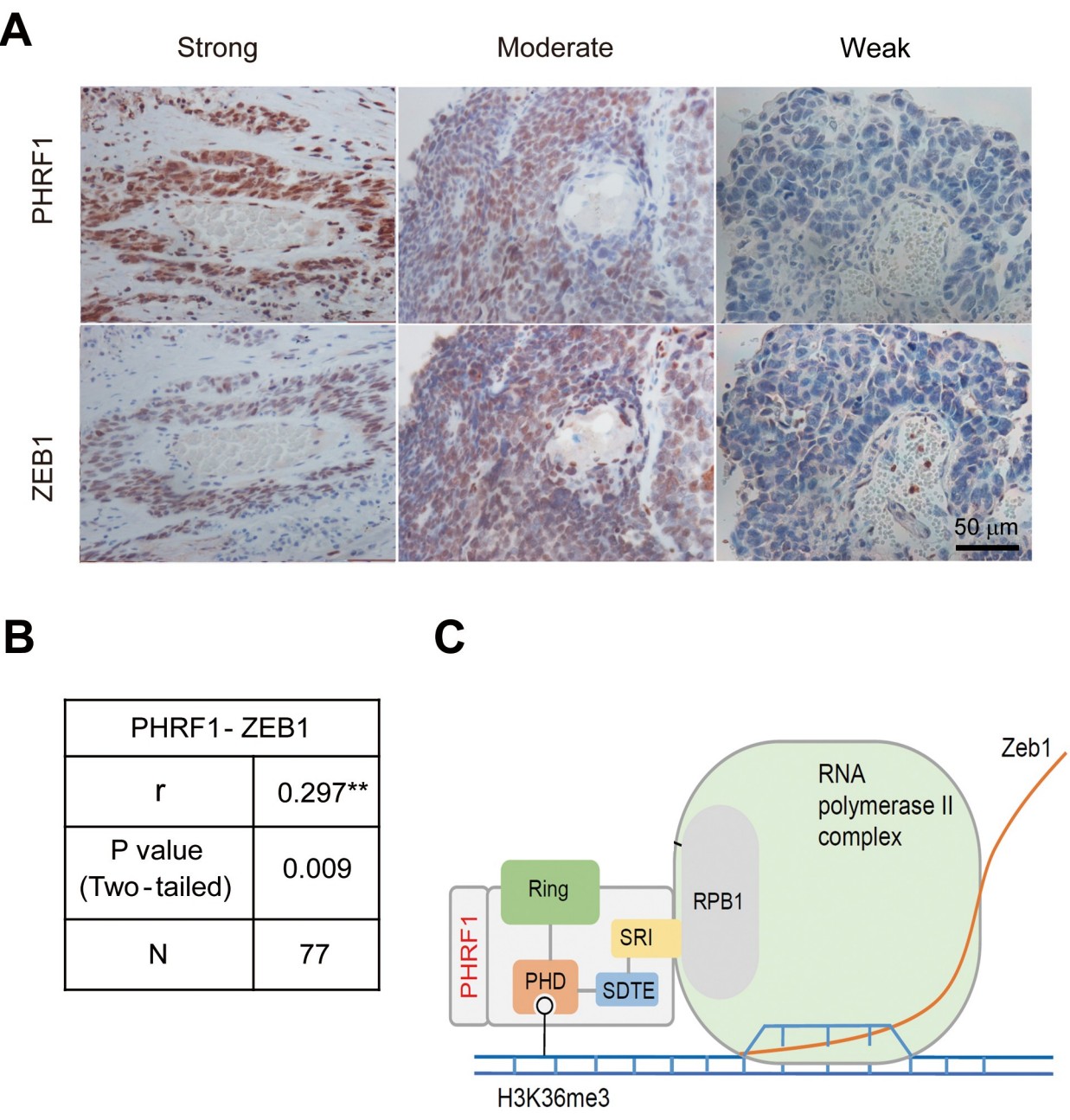

**Fig 5. Immunohistochemistry staining of PHRF1 and ZEB1 in lung cancer specimens.** (A) Lung cancer specimens were immunostained with anti-PHRF1 and anti-ZEB1 antibodies. The expression localization of PHRF1 coincided with that of ZEB1 in the same tumor specimens. Scale bar, 50 μm. (B) Spearmann's correlation analysis of PHRF1 vs. ZEB1 immunohistochemistry pairs (n = 77). (C) A schematic model for PHRF1 to modulate ZEB1's transcription.

moderately expressed specimens (n = 32), and in 80% of PHRF1 strongly expressed specimens (n = 40), [respectively,?] indicating that PHRF1 and ZEB1 exhibited a positive correlation in lung cancer specimens. We found significant and positive co-expression correlations between PHRF1 and ZEB1 (Fig 5B). Collectively, these results suggest that PHRF1 substantially modulated the expression of ZEB1, and possibly as well as ZEB2, to promote tumorigenesis.

## Discussion

On the basis of our findings, we proposed a model for PHRF1 to modulate the expression of ZEB1 and promote cancer migration and invasion. PHRF1 binds to the phosphorylated C-terminal repeat domain (CTD) of Rpb1 and is recruited to the TSS region of *ZEB1*. PHRF1 targets pVHL for degradation to maintain the stability of HIF1α/HIF2α. Both are capable of promoting the transcription of *ZEB1* (Fig 5C).

A number of reports have shown that aberrant perturbation of the epithelial-mesenchymal transition (EMT) triggers malignant tumor progression and endows tumor cells with greater motility, self-renewal capacity, resistance to drugs, and degradation of the extracellular matrix to facilitate their invasion into the surrounding tissues and eventual metastasis to distant organs [1–5]. EMT-associated transcription factors like SNAI1, Twist and ZEB1/2, have conferred considerable research highlights to promote EMT events [2]. Overexpression of ZEB1 and Zeb2 in epithelial cells have been demonstrated to induce the EMT, and ZEB1 is expressed at high levels in invading lung, uterine, colorectal, endometrial, prostate, and gallbladder carcinomas [27–32]. Our data revealed that, among the classic EMT-associated transcription factors (SNAI1 and ZEB1/2), PHRF1 substantially regulates the expression of ZEB1/2 and is required for the initiation of the EMT process in lung cancer cells. Overexpression of PHRF1 was able to induce ZEB1's expression, whereas suppression of PHRF1 impeded ZEB1 and EMT in lung cancer cells (Fig 3), most likely by a compelling involvement of PHRF1 in collaboration with Rpb1. However, we are still at an early stage to address why PHRF1 specifically occupies the transcription start site region of ZEB1/2, but does not engage with that of SNAI1. One possibility is that there are specific histone signatures in the site's region of the ZEB1 gene, such as H3K36me2/3. Nevertheless, PHRF1$^{\Delta PHD}$ did not compromise ZEB1's expression (Fig 5A), diminishing the speculation that the binding of PHRF1 with H3K36me2/3 is essential for induction of ZEB1's expression.

Phosphorylation of the CTD in Rpb1 creates a platform for the engagement of RNA processing factors and chromatin-modifying factors that facilitate RNA synthesis [33, 34]. The Set2 Rpb1 interacting domain (SRI) preferentially binds to the CTD phosphorylated at both ser2 and ser5. Deletion of the SRI domain in yeast Set2 impairs transcription elongation by RNAPII [35, 36]. Asr1, the ortholog of human PHRF1 in *S. cerevisiae*, contains a short form of PHRF1 with RING, PHD and SRI domains that bind to the phosphorylated CTD at Ser5. This binding enhances the ubiquitination of Rpb1 and Rpb2, the dissociation of Rpb4/Rpb7 and RNA polymerase inactivation in yeast [37]. Interestingly, unlike Asr1, we found that PHRF1's SRI mainly associated with the CTD phosphorylated at Ser2 and Ser5 (Fig 4C), indicating that PHRF1 may work with the RNPII complex to increase the transcription of its target genes. Indeed, *ZEB1* and *ZEB2* were unequivocally elevated in the presence of PHRF1 (Fig 4D). PHRF1$^{\Delta SRI}$ remarkably attenuated the expression level of ZEB1, rendering the possibility that the induction of ZEB1 by PHRF1 bridges the connection of PHRF1 with Rpb1.

Wang *et al*. described that PHRF1 inhibits H1299 cell proliferation, colony formation *in vitro*, and growth of tumor xenograft *in vivo* [23]. H1299 cells have a homozygous partial deletion of the *p53* gene and lack expression of p53 protein. By contrast, the A549 cell contains a wild-type *p53*, and the Cl1-0 cell has a p53R248W mutation, indicating that PHRF1 may promote or suppress cell proliferation under different genetic backgrounds. Additionally, Ettahar *et al*. identified that PHRF1 has a role as a tumor suppressor, promoting the TGF-β cytostatic program in acute promyelocytic leukemia pathogenesis [20, 21], which may be considered contradictory to our results. However, mounting evidence has shown that several factors, such as TGF-β, may play different roles in promoting cell proliferation and disseminating cancer cells to metastatic niches. If PHRF1 is such a case, it would be plausible to suggest that PHRF1

not only suppresses cell proliferation in early stage in cancer cells, but also facilitates cell migration and invasion during the later metastatic stage.

## Supporting information

**S1 Fig. PHRF1 expression is associated with overall and progression-free survival (PFS) of cancer patients.** (A) Ovarian cancer. n = 1436. (B) Gastric cancer. n = 882. (C) Lung adeno-carcinoma. n = 513. (D) Lung squamous carcinoma. n = 501. (E) Liver hepatocellular carci-noma. n = 371. (F) Cervical squamous carcinoma. n = 304. The cohorts were divided into two groups, high (red) and low (black), according to the median expression value of PHRF1, which were retrieved from the Kaplan–Meier plotter database (http://kmplot.com/analysis/index.php?p=service&cancer). (G) Lung adenocarcinoma. n = 203. (H) Cervical squamous carci-noma. n = 275. Patient data were obtained from cBioPortal TCGA Nature 2014 and TCGA PanCancer Atlas datasets, respectively. Z score $\geq$ 2 (red).
(TIF)

**S2 Fig. Clonogenic assays in PHRF1-depleted and -overexpressing lung cancer A549 cells.**
(A) Knockdown of PHRF1 with two specific siRNAs, and (B) overexpression of PHRF1 with two lentiviral transductions, were carried out in lung cancer A549 cells. Anchorage-dependent colony formation was examined for 12 days of culture and then stained with crystal violet. Col-onies larger than 0.1 mm in diameter were scored. Quantitative results are shown in the right panels. (C) Cell proliferation was measured using a BrdU colorimetric assay (Roche Diagnos-tics, Mannheim, Germany). Each bar represents the mean ± SD of three independent experi-ments. (**P < 0.01 and ***P < 0.001 compared with the controls). (D) PHRF1-depleted and -overexpressing A549 cells were subjected to wound healing assay for 24 h in 0.5% FBS culture medium. Scale bar, 0.1 mm.
(TIF)

**S3 Fig. Effects of PHRF1 on invasion in CL1-0 and CL1-5 lung cancer cells.** (A) $1\times10^4$ con-trol and PHRF1-depleted CL1-5 cells were seeded on the top well of a Boyden chamber in serum-free media, while culture medium supplemented with serum was placed in the well below for 24 h. Cells were photographed under phase-contrast microscopy and quantified. (B) $1\times10^4$ control and PHRF1-overexpressing CL1-0 cells mixed with Matrigel were placed on the top of invasive chambers and allowed to penetrate to the lower surface of the filter. The cells on the lower surface of the membrane were stained with crystal violet and photographed under a light microscope. Scale bar, 100 μm.
(TIF)

**S4 Fig. PHRF1 was not induced by HIF-1/2α.** (A) Schematic representation of the proximal promoter (~ 350 nt upstream) of the *PHRF1* gene. HRE: hypoxia response element. (B) Cell extracts prepared from HIF1α and HIF2α transfected HEK293T cells were immunoblotted with indicated antibodies. All Western blots were processed in identical conditions and cropped from S3 Raw images.
(TIF)

**S5 Fig. PHRF1 is required for pVHL ubiquitination.** (A) Control and PHRF1-depleted A549 cells were placed in the hypoxia chamber (1% $O_2$) for different time points and then immuno-blotted with indicated antibodies. (B) Control and PHRF1 aberrant mutants were transfected into A549 cells and immunoblot analysis was carried out with indicated antibodies. (C) Con-trol and PHRF1-depleted A549 cells were placed in the hypoxia chamber (1% $O_2$) for different time points and then immunoblotted with anti-pVHL antibody. (D) HA-PHRF1 and

HA-PHRF1$^{C108A}$ were co-transfected with FLAG-Ub and SBP-pVHL into HEK293T cells for 45 h and incubated with MG132 for another 3 h. Cell extracts were pulled down by streptavidin agarose and immunoblotted with indicated antibodies to detect ubiquitinated pVHL. All Western blots were processed in identical conditions and cropped from S3 Raw images.
(TIF)

**S1 Table. Primer sequences of RT-qPCR.**
(DOC)

**S2 Table. Primer sequences of ChIP.**
(DOCX)

**S1 Raw images. Uncropped images for all gels and Western blots.**
(TIF)

**S2 Raw images. Uncropped images for all gels and Western blots.**
(TIF)

**S3 Raw images. Uncropped images for all gels and Western blots.**
(TIF)

## Acknowledgments

We thank the national core facility at the Genomic Research Center, Academia Sinica, Taiwan, for RNAi reagents.

## Author Contributions

**Conceptualization:** Mau-Sun Chang.

**Data curation:** Nai-Lin Chou, Hung-Wei Lin.

**Formal analysis:** Jin-Yu Lee, Chih-Chen Fan, Nai-Lin Chou, Hung-Wei Lin.

**Funding acquisition:** Mau-Sun Chang.

**Investigation:** Jin-Yu Lee, Chih-Chen Fan, Nai-Lin Chou, Hung-Wei Lin.

**Software:** Hung-Wei Lin.

**Validation:** Jin-Yu Lee.

**Writing – original draft:** Mau-Sun Chang.

**Writing – review & editing:** Mau-Sun Chang.

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
