## [Decision Letter · Decision Letter 0]

27 May 2020

PONE-D-20-04249

PHRF1 promotes migration and invasion by modulating ZEB1 expression

PLOS ONE

Dear Dr. Chang,

Thank you for submitting your manuscript to PLOS ONE. After careful consideration, we feel that it has merit but does not fully meet PLOS ONE’s publication criteria as it currently stands. Therefore, we invite you to submit a revised version of the manuscript that addresses the points raised during the review process.

We look forward to receiving your revised manuscript.

Kind regards,

Olorunseun Ogunwobi, MD, PhD

Academic Editor

PLOS ONE

Journal Requirements:

2. At this time, we request that you  please report additional details in your Methods section regarding animal care, as per our editorial guidelines:

(1) Please state the source and number of mice used in the study  

(2) Please include the secondary and confirmatory method of euthanasia in addition to CO2 asphyxiation

(3) In Figure 2E, please provide the absolute tumor volumes (mm3 or cm3) on the y-axis, as opposed to the fold difference.

Thank you for your attention to these requests.

3. Please provide the full name of the IRB that approved the use of patient samples in this study in the ethics statement of the online submission form.

4. In the Methods section, please provide additional information about the patient samples used in your study, including:

a) the source of the tissue samples analyzed in this work (e.g. hospital, institution or medical center name).

b) the date range (month and year) during which patients whose tissue samples were selected for this study sought treatment.

5. Please provide additional information about each of the cell lines used in this work, including their history and any quality control testing procedures (authentication, characterisation, and mycoplasma testing). For more information, please see http://journals.plos.org/plosone/s/submission-guidelines#loc-cell-lines. In addition, please provide the source of the F12K, DMEM and RPMI1640 medium used.

6. Please provide the product numbers and any lot numbers of the primary anitbodies used in your study in the Methods section."

7. Please provide scale bars on the microscopy images presented in Figures 1, 2, 3 and 5 and refer to the scale bar in the corresponding Figure legends.

8. To comply with PLOS ONE submission guidelines, in your Methods section, please provide additional information regarding your statistical analyses. For more information on PLOS ONE's expectations for statistical reporting, please see https://journals.plos.org/plosone/s/submission-guidelines.#loc-statistical-reporting.

9. We note that you have included the phrase “data not shown” in your manuscript. Unfortunately, this does not meet our data sharing requirements. PLOS does not permit references to inaccessible data. We require that authors provide all relevant data within the paper, Supporting Information files, or in an acceptable, public repository. Please add a citation to support this phrase or upload the data that corresponds with these findings to a stable repository (such as Figshare or Dryad) and provide and URLs, DOIs, or accession numbers that may be used to access these data. Or, if the data are not a core part of the research being presented in your study, we ask that you remove the phrase that refers to these data.

10. PLOS requires an ORCID iD for the corresponding author in Editorial Manager on papers submitted after December 6th, 2016. Please ensure that you have an ORCID iD and that it is validated in Editorial Manager. To do this, go to ‘Update my Information’ (in the upper left-hand corner of the main menu), and click on the Fetch/Validate link next to the ORCID field. This will take you to the ORCID site and allow you to create a new iD or authenticate a pre-existing iD in Editorial Manager. Please see the following video for instructions on linking an ORCID iD to your Editorial Manager account: https://www.youtube.com/watch?v=_xcclfuvtxQ

11. PLOS ONE now requires that authors provide the original uncropped and unadjusted images underlying all blot or gel results reported in a submission’s figures or Supporting Information files. This policy and the journal’s other requirements for blot/gel reporting and figure preparation are described in detail at https://journals.plos.org/plosone/s/figures#loc-blot-and-gel-reporting-requirements and https://journals.plos.org/plosone/s/figures#loc-preparing-figures-from-image-files. When you submit your revised manuscript, please ensure that your figures adhere fully to these guidelines and provide the original underlying images for all blot or gel data reported in your submission. See the following link for instructions on providing the original image data: https://journals.plos.org/plosone/s/figures#loc-original-images-for-blots-and-gels.

13. Thank you for stating the following financial disclosure:

'No'

14. Thank you for stating the following in your Competing Interests section: 

'No'

Additional Editor Comments (if provided):

Reviewers' comments:

Reviewer's Responses to Questions

**Comments to the Author**

1. Is the manuscript technically sound, and do the data support the conclusions?

Reviewer #1: Yes

Reviewer #2: Yes

2. Has the statistical analysis been performed appropriately and rigorously? 

Reviewer #1: Yes

Reviewer #2: Yes

3. Have the authors made all data underlying the findings in their manuscript fully available?

Reviewer #1: Yes

Reviewer #2: Yes

4. Is the manuscript presented in an intelligible fashion and written in standard English?

Reviewer #1: No

Reviewer #2: Yes

5. Review Comments to the Author

Reviewer #1: The ms by Lee and colleagues entitled “PHRF1 promotes migration and invasion by modulating ZEB1 expression” describes a novel mechanism of ZEB1 regulation by PHRF1, a protein with an important role in APL tumorigenesis, contributing to EMT process in lung cancer cells. The authors have checked the levels of PHRF1 in different tumors using a publicly accessible database and have associated them with overall and progression free survival. Furthermore, authors have studied the effect of PHRF1 silencing and PHRF1 overexpression on cell migration and invasion, as well as on the levels of EMT markers, such as E-cadherin and N-cadherin suggesting an important role of this protein in the lung cells malignant transformation. Authors also confirmed that ZEB1 is responsible for PHRF1-mediated migration and invasion analyzing the silencing of ZEB1 in PHRF1- overexpressing lung cells. Finally, direct PHRF1 binding adjacent to the TSS region of ZEB1 was confirmed in ChIP-PCR. In summary, data presented by Lee and colleagues suggest that PHRF1 could modulate the expression of ZEB1 promoting migration and invasion in cancer lung cells.

Overall, the data in this ms are novel and important, nevertheless I see several defficiencies that should be addressed:

Major comments:

1. Abstract and Introduction: I suggest to rewrite all and I recommend a revision by native speaker. There are some nonsense sentences as in line 25 "The C-terminal Set2 Rpb1 Interacting (SRI) domain…”, I really do not understand what you are trying to mean. Probably, it is incorrectly written, so please, rewrite all trying to make sense of the text and with a clear and concise message.

I feel the same in the Introduction section where there are a huge amount of irrelevant information and where it is impossible to get any conclusion (lines 55 to 66).

2. The sequences of primers you listed in S1 table are wrong; they have no targets or they match with predicted targets that are not the genes that supposed to be. Please, recheck all the sequences.

3. Have the authors blocked proliferation when performing migration assays? In this sense, I think 0’5% of FBS is a better way to avoid proliferation in this kind of experiments.

4. In results the authors resume the association of PHRF1 expression with overall and progression free survival in a number human tumors that are so far in publicly database. Regarding this point it would be interesting to do an analysis of expression in TCGA datasets using tools like cbioportal, firebrowse, etc. I also suggest, using the TCGA database and the available online tools, to do perform an analysis of whether the expression of this protein correlates with the clinical attributes of the tumor. This point could improve the study and could support the data presented in the manuscript.

5. In Western blots reported in Figure 1E, densitometry with statistical analysis must be added. Regarding this point, it is remarkable that control cells in silencing experiments appear to have reasonable PHRF1 protein expression in comparison to control cells in overexpressing experiments. Do the authors have an explanation for this differences in the expression? Finally, I missed the N-cadherin protein expression in the silencing experiments. Did you analyze N-cadherin expression in the PHRF1 silenced cells?

6. In western blots reported in Figures 3A and 3B please add densitometry with statistical analysis.

7.Fig 4A. Densitometry with statistical analysis must be added.

8. ChIP-PCR (Fig. 4E): an important negative control in ChIP experiments is a distant region in cis to the putative binding site (e.g. a kb away from the promoter). Could this be included?

10. You cite a paper of Wang et al. (2016) in the Introduction demonstrating a tumor suppressor role of PHRF1 in lung cancer, however, I am surprised that these results are not discussed in the Discussion section. How do you explain these controversial data?

Minor comments:

1. Use capital letters for human genes: ZEB2 in lines: 250, 252, 253, 254, 291, 303 and 323.

2. All human genes have to be in italics.

3. In line 269: “PHRF1 and ZEB1 were induced in control cells under hypoxia; nonetheless, the amount of ZEB1 was unable to increase in PHRF1 depleted cells under low O2 condition (S5A Fig).”; text does not correspond with the figure, and the figure S5A is missing.

Reviewer #2: The authors of this paper carried out a very interesting investigation of the implications behind PHRF1 potentially acting as a promoter of migration and invasion via ZEB1 expression. They used a variety of well thought-out experiments to show this, including overexpressing and knocking-down expression of PHRF1 in lung cancer cells and seeing their effects on Zeb1 and migration and invasion, among other factors implying tumorigenicity. They also carry out ChIP in order to see binding of PHRF1 to facilitate RNA Pol 2 transcription/elongation to further strengthen their claim that PHRF1 may be capable of regulating the trx of genes like Zeb1.

While the study is a very fascinating one, please consider the following suggestions.

1) There are several English expression errors throughout the paper. Please revise and make sure to correct.

2) While this paper is full of different molecules and the authors generally do a good job at defining each molecule before using the abbreviation, they don’t do so for PHRF1. Please do so. And also NBS1. What does this stand for? Those were 2 molecules that I noticed but please go over to make sure haven’t missed any others.

3) For both the Abstract and the Introduction, please make a better connection between what is known about PHRF1 and your interests for studying it as a promoter of tumor invasion/migration/progression. Ettahar A et al 2013 “Identification of PHRF1 as a tumor suppressor that promotes the TGFB cytostatic program through selective release…” seems to suggest PHRF1 may work as the exact opposite of what this paper is seeing. Please explain this conflicting observation. What is known about PHRF1? Is work the first to suggest PHRF1 may actually not be a tumor suppressor? What have others published?

4) For Figure 1E, this piece of data especially for E-cadherin is ok but not the most convincing. Have you tried looking at other EMT markers perhaps? To show EMT is occurring?

6. PLOS authors have the option to publish the peer review history of their article (what does this mean?). If published, this will include your full peer review and any attached files.

Reviewer #1: No

Reviewer #2: No

---

## [Author Response · Author response to Decision Letter 0]

12 Jun 2020

Q1. Please ensure that your manuscript meets PLOS ONE's style requirements, including those for file naming. The PLOS ONE style templates can be found at https://journals.plos.org/plosone/s/file?id=wjVg/PLOSOne_formatting_sample_main_body.pdf and https://journals.plos.org/plosone/s/file?id=ba62/PLOSOne_formatting_sample_title_authors_affiliations.pdf.

R1. We have followed the style templates to reformat our manuscript.

Q2. At this time, we request that you please report additional details in your Methods section regarding animal care, as per our editorial guidelines:

(1) Please state the source and number of mice used in the study 

(2) Please include the secondary and confirmatory method of euthanasia in addition to CO2 asphyxiation

(3) In Figure 2E, please provide the absolute tumor volumes (mm3 or cm3) on the y-axis, as opposed to the fold difference.

R2. (1) Three mice of each group were purchased from BioLASCO (Taipei, Taiwan), which is stated in line 154.

 (2) To prevent a possibility that mice might recover from a deep CO2 exposure, a confirmatory means of euthanasia, such as cervical dislocation or 50 % additional time in the euthanasia chamber, was conducted, which is stated in line 158.

 (3) The absolute tumor volumes (cm3) is added on the y-axis.

Q3. Please provide the full name of the IRB that approved the use of patient samples in this study in the ethics statement of the online submission form.

R3. A study regarding “the expression of PHRF1 in the lung carcinoma” was approved by the Institutional Review Board of Mackay Memorial Hospital (IRB No: 15MMHIS012), which is stated in line 169.

Q4. In the Methods section, please provide additional information about the patient samples used in your study, including:

a) the source of the tissue samples analyzed in this work (e.g. hospital, institution or medical center name).

b) the date range (month and year) during which patients whose tissue samples were selected for this study sought treatment.

R4. We have the following statement in line 172, “Tissue specimens of 80 patients with lung cancer carcinomas, ranging from January 2008 to June 2014, were selected for IHC analyses based on availability of archival human lung tissue blocks from diagnostic inspections by pathologists at MacKay Memorial Hospital, Taipei, Taiwan” 

Q5. Please provide additional information about each of the cell lines used in this work, including their history and any quality control testing procedures (authentication, characterisation, and mycoplasma testing). For more information, please see http://journals.plos.org/plosone/s/submission-guidelines#loc-cell-lines. In addition, please provide the source of the F12K, DMEM and RPMI1640 medium used. 

R5. A549 human lung cancer cells obtained from the American Type Culture Collection (ATCC; Rockville, MD) and cultured in F12K medium (Hyclone, Utah, USA) supplemented with 10% FBS. HEK293T cells were obtained from ATCC and maintained in DMEM medium (Hyclone, Utah, USA) with 10% FBS. Human lung adenocarcinoma cells CL1-0 and CL1-5 were established at The National Taiwan University College of Medicine and maintained in RPMI1640 medium (Hyclone, Utah, USA) with 10% FBS. All cell lines were submitted to real time PCR for mycoplasma detection and short tandem repeats identification by capillary electrophoresis for cell line authentication, which is stated in line 98.

Q6. Please provide the product numbers and any lot numbers of the primary anitbodies used in your study in the Methods section."

R6. We have provided the product numbers of the primary antibodies in the Methods section.

Q7. Please provide scale bars on the microscopy images presented in Figures 1, 2, 3 and 5 and refer to the scale bar in the corresponding Figure legends.

R7. We have provided scale bars in Figures 1, 2, 3 and 5 and refer to the scale bar in the corresponding Figure legends. 

Q8. To comply with PLOS ONE submission guidelines, in your Methods section, please provide additional information regarding your statistical analyses. For more information on PLOS ONE's expectations for statistical reporting, please see https://journals.plos.org/plosone/s/submission-guidelines.#loc-statistical-reporting.

R8. We have provided statistical analyses in Methods section in line 182. 

Q9. We note that you have included the phrase “data not shown” in your manuscript. Unfortunately, this does not meet our data sharing requirements. PLOS does not permit references to inaccessible data. We require that authors provide all relevant data within the paper, Supporting Information files, or in an acceptable, public repository. Please add a citation to support this phrase or upload the data that corresponds with these findings to a stable repository (such as Figshare or Dryad) and provide and URLs, DOIs, or accession numbers that may be used to access these data. Or, if the data are not a core part of the research being presented in your study, we ask that you remove the phrase that refers to these data.

R9. Thanks for this suggestion. We have removed the phrase “data not shown”.

Q10. PLOS requires an ORCID iD for the corresponding author in Editorial Manager on papers submitted after December 6th, 2016. Please ensure that you have an ORCID iD and that it is validated in Editorial Manager. To do this, go to ‘Update my Information’ (in the upper left-hand corner of the main menu), and click on the Fetch/Validate link next to the ORCID field. This will take you to the ORCID site and allow you to create a new iD or authenticate a pre-existing iD in Editorial Manager. Please see the following video for instructions on linking an ORCID iD to your Editorial Manager account: https://www.youtube.com/watch?v=_xcclfuvtxQ

R10. I have validated my ORCID account in Editorial Manager.

Q11. PLOS ONE now requires that authors provide the original uncropped and unadjusted images underlying all blot or gel results reported in a submission’s figures or Supporting Information files. This policy and the journal’s other requirements for blot/gel reporting and figure preparation are described in detail at https://journals.plos.org/plosone/s/figures#loc-blot-and-gel-reporting-requirements and https://journals.plos.org/plosone/s/figures#loc-preparing-figures-from-image-files. When you submit your revised manuscript, please ensure that your figures adhere fully to these guidelines and provide the original underlying images for all blot or gel data reported in your submission. See the following link for instructions on providing the original image data: https://journals.plos.org/plosone/s/figures#loc-original-images-for-blots-and-gels.

R11. We have provided the original uncropped and unadjusted images underlying all blot or gel results reported in a submission’s figures or Supporting Information files.

Q12. In your cover letter, please note whether your blot/gel image data are in Supporting Information or posted at a public data repository, provide the repository URL if relevant, and provide specific details as to which raw blot/gel images, if any, are not available. Email us at plosone@plos.org if you have any questions.

R12. We have stated in our cover letter that blot/gel image data are in Supporting Information.

Q13. Thank you for stating the following financial disclosure:

'No'

a. Please clarify the sources of funding (financial or material support) for your study. List the grants or organizations that supported your study, including funding received from your institution.

d. If you did not receive any funding for this study, please state: “The authors received no specific funding for this work.”

R13. a. This work was supported by Ministry of Science and Technology (MOST 106-2311-B-002-004).

 c. All authors did not receive a salary from our funders.

 d. We will include the amended statements in the cover letter. 

Q14. Thank you for stating the following in your Competing Interests section: 

'No'

R14. The authors have declared that no competing interests exist. This information should be included in our cover letter

Reviewer's Responses to Questions

Comments to the Author

Q4. Is the manuscript presented in an intelligible fashion and written in standard English? PLOS ONE does not copyedit accepted manuscripts, so the language in submitted articles must be clear, correct, and unambiguous. Any typographical or grammatical errors should be corrected at revision, so please note any specific errors here.

Reviewer #1: No

Reviewer #2: Yes

 R4. An independent editor, Marc Anthony, has assisted us to correct English grammar and spelling mistakes.

Reviewer #1: 

Major comments:

Q1. Abstract and Introduction: I suggest to rewrite all and I recommend a revision by native speaker. There are some nonsense sentences as in line 25 "The C-terminal Set2 Rpb1 Interacting (SRI) domain…”, I really do not understand what you are trying to mean. Probably, it is incorrectly written, so please, rewrite all trying to make sense of the text and with a clear and concise message.

I feel the same in the Introduction section where there are a huge amount of irrelevant information and where it is impossible to get any conclusion (lines 55 to 66).

R1. Thanks for this suggestion. An independent editor, Marc Anthony, has assisted us to correct English grammar and spelling mistakes, especially in the abstract and introduction section. 

Q2. The sequences of primers you listed in S1 table are wrong; they have no targets or they match with predicted targets that are not the genes that supposed to be. Please, recheck all the sequences.

R2. Thanks for this reminder. We mixed S1 table with S2 table. This mistake has been corrected.

Q3. Have the authors blocked proliferation when performing migration assays? In this sense, I think 0.5 % of FBS is a better way to avoid proliferation in this kind of experiments. 

R3. To clarify whether proliferation rate affected the cell migration, PHRF1-depleted and -overexpressing A549 cells were cultured in 0.5 % of FBS and wound healing assays were conducted for 24 hr. PHRF1 knockdown significantly reduced the motility, but PHRF1 overexpression had little effect on cell migration compared with controls in in 0.5 % of FBS medium (S2D Fig), which is stated in line 252.

Q4. In results the authors resume the association of PHRF1 expression with overall and progression free survival in a number human tumors that are so far in publicly database. Regarding this point it would be interesting to do an analysis of expression in TCGA datasets using tools like cbioportal, firebrowse, etc. I also suggest, using the TCGA database and the available online tools, to do perform an analysis of whether the expression of this protein correlates with the clinical attributes of the tumor. This point could improve the study and could support the data presented in the manuscript.

R4. Thanks for this suggestion. An elevated PHRF1 expression is associated with poor survival in the lung adenocarcima and cervical squamous cell carcinoma using the cBioportal TCGA dataset analysis (S1G-H Fig), which is stated in line 197.

Q5. In Western blots reported in Figure 1E, densitometry with statistical analysis must be added. Regarding this point, it is remarkable that control cells in silencing experiments appear to have reasonable PHRF1 protein expression in comparison to control cells in overexpressing experiments. Do the authors have an explanation for this differences in the expression? Finally, I missed the N-cadherin protein expression in the silencing experiments. Did you analyze N-cadherin expression in the PHRF1 silenced cells?

R5. a. The densitometry result is added to Fig 1E.

 b. PHRF1 expression in silencing cells was over-exposed. We have replaced it with a short-exposure result, which is in line with the uncropped image from S1_raw figure.

c. Thanks for the suggestion. We have added fibronectin in PHRF1-overexpressing cells and N-cadherin and vimentin in PHRF1-silenced cells to Fig 1E. 

Q6. In western blots reported in Figures 3A and 3B please add densitometry with statistical analysis.

R6. Densitometry results are added to Fig 3A and 3B.

Q7. Fig 4A. Densitometry with statistical analysis must be added.

R7. Densitometry results are added to Fig 3A and 3B.

Q8. ChIP-PCR (Fig. 4E): an important negative control in ChIP experiments is a distant region in cis to the putative binding site (e.g. a kb away from the promoter). Could this be included?

R8. We have included 1 kb away from the ZEB1 and ZEB2 TSSs as negative controls in Fig 4E and stated in line 299.

Q9. You cite a paper of Wang et al. (2016) in the Introduction demonstrating a tumor suppressor role of PHRF1 in lung cancer, however, I am surprised that these results are not discussed in the Discussion section. How do you explain these controversial data.

R9. Thanks to raise this question. We have the following statement, starting from line 400. “Wang et al. described that PHRF1 inhibits H1299 cell proliferation, colony formation in vitro, and growth of tumor xenograft in vivo [23]. H1299 cells have a homozygous partial deletion of the p53 gene and lack expression of p53 protein. By contrast, the A549 cell contains a wild-type p53, and the Cl1-0 cell has a p53R248W mutation, indicating that PHRF1 may promote or suppress cell proliferation under different genetic backgrounds. Additionally, Ettahar et al. identified that PHRF1 has a role as a tumor suppressor, promoting the TGF-� cytostatic program in acute promyelocytic leukemia pathogenesis [20,21], which may be considered contradictory to our results. However, mounting evidence has shown that several factors, such as TGF-�, may play different roles in promoting cell proliferation and disseminating cancer cells to metastatic niches. If PHRF1 is such a case, it would be plausible to suggest that PHRF1 not only suppresses cell proliferation in early stage in cancer cells, but also facilitates cell migration and invasion during the later metastatic stage.”

Minor comments:

Q1. Use capital letters for human genes: ZEB2 in lines: 250, 252, 253, 254, 291, 303 and 323.

R1. We have changed capital letters for ZEB2. 

Q2. All human genes have to be in italics.

R2. We have changed human genes to be in italics. 

Q3. In line 269: “PHRF1 and ZEB1 were induced in control cells under hypoxia; nonetheless, the amount of ZEB1 was unable to increase in PHRF1 depleted cells under low O2 condition (S5A Fig).”; text does not correspond with the figure, and the figure S5A is missing.

R3. Thanks for this reminder. We correct this mistake. S5A Fig has been integrated into S5 Figure.

Reviewer #2: 

Q1. There are several English expression errors throughout the paper. Please revise and make sure to correct.

R1. An independent editor, Marc Anthony, has assisted us to correct English grammar and spelling mistakes.

Q2. While this paper is full of different molecules and the authors generally do a good job at defining each molecule before using the abbreviation, they don’t do so for PHRF1. Please do so. And also NBS1. What does this stand for? Those were 2 molecules that I noticed but please go over to make sure haven’t missed any others. 

R2. Thanks for this reminder. PHRF1 (PHD and RING finger domain-containing protein 1) in line 20. ZEB1/2 (Zinc finger E-box-binding homeobox 1/2) in line 44. NBS1 (the Nijmegen breakage syndrome gene) in line 76. 

Q3. For both the Abstract and the Introduction, please make a better connection between what is known about PHRF1 and your interests for studying it as a promoter of tumor invasion/migration/progression. Ettahar A et al 2013 “Identification of PHRF1 as a tumor suppressor that promotes the TGFB cytostatic program through selective release…” seems to suggest PHRF1 may work as the exact opposite of what this paper is seeing. Please explain this conflicting observation. What is known about PHRF1? Is work the first to suggest PHRF1 may actually not be a tumor suppressor? What have others published?

R3. A. We have tried to reorganize the abstract and introduction to have a better connection between PHRF1 and tumor invasion via this assistance of Marc Anthony.

B. We stated in line 404, “Ettahar et al. identified that PHRF1 has a role as a tumor suppressor, promoting the TGF-� cytostatic program in acute promyelocytic leukemia pathogenesis [20,21], which may be considered contradictory to our results. However, mounting evidence has shown that several factors, such as TGF-�, may play different roles in promoting cell proliferation and disseminating cancer cells to metastatic niches. If PHRF1 is such a case, it would be plausible to suggest that PHRF1 not only suppresses cell proliferation in early stage in cancer cells, but also facilitates cell migration and invasion during the later metastatic stage.”

Q4. For Figure 1E, this piece of data especially for E-cadherin is ok but not the most convincing. Have you tried looking at other EMT markers perhaps? To show EMT is occurring?

R4. Thanks for the suggestion. We have added fibronectin in PHRF1-overexpressing cells and N-cadherin and vimentin in PHRF1-silenced cells to Fig 1E.

---

## [Decision Letter · Decision Letter 1]

16 Jul 2020

PHRF1 promotes migration and invasion by modulating ZEB1 expression

PONE-D-20-04249R1

Dear Dr. Mau-Sun Chang,

We’re pleased to inform you that your manuscript has been judged scientifically suitable for publication and will be formally accepted for publication once it meets all outstanding technical requirements.

Kind regards,

Olorunseun Ogunwobi, MD, PhD

Academic Editor

PLOS ONE

Reviewers' comments:

Reviewer's Responses to Questions

**Comments to the Author**

1. If the authors have adequately addressed your comments raised in a previous round of review and you feel that this manuscript is now acceptable for publication, you may indicate that here to bypass the “Comments to the Author” section, enter your conflict of interest statement in the “Confidential to Editor” section, and submit your "Accept" recommendation.

Reviewer #1: All comments have been addressed

2. Is the manuscript technically sound, and do the data support the conclusions?

Reviewer #1: Yes

3. Has the statistical analysis been performed appropriately and rigorously? 

Reviewer #1: Yes

4. Have the authors made all data underlying the findings in their manuscript fully available?

Reviewer #1: Yes

5. Is the manuscript presented in an intelligible fashion and written in standard English?

Reviewer #1: Yes

6. Review Comments to the Author

Reviewer #1: (No Response)

7. PLOS authors have the option to publish the peer review history of their article (what does this mean?). If published, this will include your full peer review and any attached files.

Reviewer #1: No

---

## [Editor Report · Acceptance letter]

20 Jul 2020

PONE-D-20-04249R1 

PHRF1 promotes migration and invasion by modulating ZEB1 expression 

Dear Dr. Chang:

I'm pleased to inform you that your manuscript has been deemed suitable for publication in PLOS ONE. Congratulations! Your manuscript is now with our production department. 

Kind regards, 

on behalf of

Dr Olorunseun Ogunwobi 

Academic Editor

PLOS ONE